# Next-Generation Sequencing Technology: Current Trends and Advancements

**DOI:** 10.3390/biology12070997

**Published:** 2023-07-13

**Authors:** Heena Satam, Kandarp Joshi, Upasana Mangrolia, Sanober Waghoo, Gulnaz Zaidi, Shravani Rawool, Ritesh P. Thakare, Shahid Banday, Alok K. Mishra, Gautam Das, Sunil K. Malonia

**Affiliations:** 1miBiome Therapeutics, Mumbai 400102, India; heena@mibiome.com (H.S.); kandarpbioinfo@gmail.com (K.J.); upasana@microbiome.in (U.M.); sanober@microbiome.in (S.W.); gulnaz@microbiome.in (G.Z.); shravani@mibiome.com (S.R.); 2Department of Molecular Cell and Cancer Biology, UMass Chan Medical School, Worcester, MA 01605, USA; ritesh.thakare@umassmed.edu (R.P.T.); shahidkhursheed.banday@umassmed.edu (S.B.); alok.mishra@umassmed.edu (A.K.M.)

**Keywords:** next-generation sequencing, genomics, microbiome, molecular diagnostics, bioinformatics, Nanopore, PacBio, Illumina, pyrosequencing

## Abstract

**Simple Summary:**

Next-generation sequencing (NGS) is a powerful tool used in genomics research. NGS can sequence millions of DNA fragments at once, providing detailed information about the structure of genomes, genetic variations, gene activity, and changes in gene behavior. Recent advancements have focused on faster and more accurate sequencing, reduced costs, and improved data analysis. These advancements hold great promise for unlocking new insights into genomics and improving our understanding of diseases and personalized healthcare. This review article provides an overview of NGS technology and its impact on various areas of research, such as clinical genomics, cancer, infectious diseases, and the study of the microbiome.

**Abstract:**

The advent of next-generation sequencing (NGS) has brought about a paradigm shift in genomics research, offering unparalleled capabilities for analyzing DNA and RNA molecules in a high-throughput and cost-effective manner. This transformative technology has swiftly propelled genomics advancements across diverse domains. NGS allows for the rapid sequencing of millions of DNA fragments simultaneously, providing comprehensive insights into genome structure, genetic variations, gene expression profiles, and epigenetic modifications. The versatility of NGS platforms has expanded the scope of genomics research, facilitating studies on rare genetic diseases, cancer genomics, microbiome analysis, infectious diseases, and population genetics. Moreover, NGS has enabled the development of targeted therapies, precision medicine approaches, and improved diagnostic methods. This review provides an insightful overview of the current trends and recent advancements in NGS technology, highlighting its potential impact on diverse areas of genomic research. Moreover, the review delves into the challenges encountered and future directions of NGS technology, including endeavors to enhance the accuracy and sensitivity of sequencing data, the development of novel algorithms for data analysis, and the pursuit of more efficient, scalable, and cost-effective solutions that lie ahead.

## 1. Introduction

Next-generation sequencing (NGS) has revolutionized genomics, expanding our knowledge of genome structure, function, and dynamics. This groundbreaking technology has enabled extensive research and allowed scientists to explore the complexities of genetic information in unprecedented ways. With its high-throughput capacity and cost-effectiveness, NGS has become a fundamental tool for researchers across diverse disciplines, from basic biology to clinical diagnostics [1]. NGS has not only enabled comprehensive genome sequencing but also facilitated transcriptomics, epigenomics, metagenomics, and other omics studies [2]. The advent of advanced NGS platforms, such as Illumina, Pacific Biosciences, and Oxford Nanopore, has transformed the field of genomics by allowing for the parallel sequencing of millions to billions of DNA fragments [3,4]. This capability has unlocked new opportunities for understanding genetic variation, gene expression, epigenetic modifications, and microbial diversity. NGS has been instrumental in identifying disease-causing variants, uncovering novel drug targets, and shedding light on complex biological phenomena, including the heterogeneity of tumors and developmental processes [3,4,5]. This review provides a comprehensive overview of NGS technology, highlighting its transformative impact in various fields, including clinical genomics, cancer research, infectious disease, surveillance, and microbiome analysis. We also discuss the future prospects of NGS, including emerging technologies, its potential for advancing genomics research, and its applications in the biomedical sciences.

## 2. Generations of Sequencing Technologies

Technologies for “reading” DNA sequences have evolved rapidly over the past two decades [6,7,8,9,10]. This rapid progress has paved the way for significant breakthroughs in the field of DNA sequencing, leading to the emergence of three generations of sequencing technologies (Figure 1).

### 2.1. First-Generation Sequencing Technology

The first attempts at sequencing DNA and RNA involved chemical degradation or enzymatic cleavage of the molecules to generate fragments that could be analyzed individually. Robert Holley was the first to sequence a nucleic acid molecule, Alanine tRNA, in 1964 using ribonuclease from *S. cerevisiae* [11]. Similarly, Walter Gilbert and Allan Maxam developed a chemical degradation technique that allowed the sequencing of complete bacteriophage PhiX174 [12]. However, the real breakthrough came with the introduction of the chain termination-based sequencing method by Fredrick Sanger [13]. This technique used dideoxynucleotides, which terminate the chain elongation of DNA strands during replication, and allowed for the production of sequence reads of up to a few hundred nucleotides in length. Sanger’s method was widely adopted and revolutionized the field of molecular biology by allowing for the rapid sequencing of DNA and RNA [12]. In 1987, the first commercial automated sequencing machine, the Applied Biosystems ABI 370, was launched in the United States. This machine used fluorescently labeled dideoxynucleotides and capillary electrophoresis to automate the Sanger sequencing method, significantly increasing the speed and accuracy of DNA sequencing [14,15]. The ABI 370 quickly became the industry standard, and subsequent improvements in the technology led to the development of higher-throughput sequencers capable of producing longer reads [15,16]. While the first-generation technology has been largely superseded by newer, higher-throughput sequencing technologies, it remains an important historical milestone in the development of sequencing techniques. The ability to sequence DNA and RNA has revolutionized many areas of biology and medicine and has led to numerous discoveries and advancements in the understanding of genetics and molecular biology.

### 2.2. Second-Generation Sequencing Technologies

Second-generation sequencing methods have revolutionized DNA sequencing by enabling the simultaneous sequencing of thousands to millions of DNA fragments. These methods differ from traditional Sanger sequencing in their ability to perform parallel sequencing. Several widely used second-generation sequencing platforms have emerged, one of which is Roche’s 454 sequencing method, which relies on pyrosequencing, where the sequence is determined by detecting the release of pyrophosphate when nucleotides are added to the DNA template. Another platform is Ion Torrent sequencing, which detects the release of hydrogen ions during DNA synthesis to determine the sequence. The widely used Illumina sequencing platform utilizes a sequencing-by-synthesis method based on reversible dye terminators. Another upcoming technology, SOLiD sequencing (Sequencing by Oligonucleotide Ligation and Detection), employs a ligation-based approach using reversible terminators to determine the DNA sequence. These second-generation sequencing technologies have significantly increased the throughput and speed of DNA sequencing, enabling a wide range of applications in genomics research and clinical diagnostics [17]. These platforms have enabled whole-genome sequencing, transcriptome analysis, and targeted sequencing, leading to breakthroughs in genetic variation, disease research, and personalized medicine. Many developments in the second generation of sequencing methods have been achieved over the years and are represented in Figure 2 and briefly described in Table 1.

### 2.3. Third-Generation Sequencing

Third-generation sequencing technologies represent the latest advancements in DNA sequencing, offering new approaches that overcome the limitations of previous generations. These technologies provide long-read sequencing capabilities, enabling the sequencing of much larger DNA fragments compared to earlier methods. Examples include PacBio Sequencing, which uses a single-molecule, real-time (SMRT) approach with fluorescently labeled nucleotides, enabling long-read sequencing of DNA fragments up to tens of kilobases in length. Another technology is Oxford Nanopore sequencing, based on nanopore technology, where a single-stranded DNA molecule passes through a nanopore, and changes in electrical current are measured to determine the DNA sequence. Oxford Nanopore sequencing provides long-read lengths, portability, and real-time analysis. Third-generation sequencing methods have been summarized in Table 1. Figure 3 describes technologies available on NGS and the type of data generated in each type of NGS assay and their brief application.

#### Long-Read and Short-Read Sequencing

The basic principle for short-read sequencing involves sequencing by synthesis based on enrichment through hybridization, amplification, or fragmentation. Whereas long-read sequencing works on sequence detection either by synthesis or by electrical voltage change/impedance, generating the current as a single base is passed through the biological membrane pore. Long-read sequencing can generate reads up to 25–30 kb, whereas short-read sequencing can generate reads around 600–700 bp. Furthermore, the amplification bias is eliminated in long-read sequencing as opposed to short-read sequencing. As the library preparation is PCR-free, the base modification such as DNA methylation can be easily detected by long-read sequencing. The introduction of high-throughput sequencing platforms has significantly reduced error rates and notably improved the accuracy of long-read sequencing technologies [29,31]. Short-read sequencing is useful for determining the abundance of specific sequences, profiling transcript expression, and identifying variants. However, long-read sequencing technologies excel in providing comprehensive genome coverage, enabling researchers to identify complex structural variants such as large insertions, deletions, inversions, duplications, and more [8,29,31].

## 3. Next-Generation Sequencing-Based Omics

Understanding complex human diseases requires data integration from multiple omics techniques such as genomics, transcriptomics, epigenomics, and proteomics. Here, we briefly describe various omics technologies that are implemented on the NGS platform:

### 3.1. Genomics

Genomics studies using NGS profoundly analyze DNA using various approaches such as whole-genome sequencing, whole-exome sequencing, and targeted sequencing.

#### 3.1.1. Whole-Genome Sequencing

Whole-genome sequencing (WGS) is a powerful and comprehensive genomic analysis technique that involves determining the complete DNA sequence of an individual’s genome. It provides a detailed blueprint of an individual’s genetic makeup, encompassing all the genes, regulatory regions, and non-coding elements present in their genome. It finds its application mainly in discovery science, such as plant and animal research, cancer research, rare genetic diseases, patients with complex disease symptoms, population genetics, and novel genome assembly of eukaryotes and prokaryotes [32]. By sequencing all the DNA in an organism’s genome, WGS enables the identification of genetic variations, ranging from single-nucleotide polymorphisms (SNPs) to larger structural changes such as insertions, deletions, and rearrangements. This wealth of information obtained through WGS offers a multitude of applications in various fields [33]. WGS has two types of sequencing approaches on the basis of genome size viz. (1) large whole-genome sequencing deciphering larger genomes of >5 Mb such as eukaryotes, and (2) small whole-genome sequencing deciphering smaller genomes of <5 Mb mainly of prokaryotes. Short-read sequencing is preferred for mutation calling, while long-read sequencing is preferred for genome assemblies. Combining short and long-read sequencing for sequencing novel genomes has been successfully applied for accurate genome assembly without a reference sequence.

#### 3.1.2. Whole-Exome Sequencing

Whole-exome sequencing (WES) is a sequencing approach that focuses on capturing and sequencing the protein-coding regions of the genome, known as the exome. The exome represents approximately 1–2% of the entire genome but contains the majority of known disease-causing variants. By sequencing the exome, WES enables the identification of genetic variations, including single-nucleotide variants (SNVs), insertions, deletions, and copy number variations (CNVs), within protein-coding genes [34,35]. WES is a cost-effective alternative to WGS for rare clinical diseases with clusters of symptoms, as well as in identifying variants for population and cancer genetics [36]. WES involves the enrichment of exonic regions using hybrid capture or target-specific amplification techniques, followed by high-throughput sequencing. Various exome capture assays from NimbleGen, Agilent, Illumina, Twist, and IDT are available that are compatible with the Illumina NGS platform [37]. The bioinformatic approach used for WES data analysis is the same as that of WGS since WES is a part of WGS.

#### 3.1.3. Targeted Sequencing

Targeted sequencing, as the name suggests, has less exploratory power than WGS or WES as it targets specific regions of the gene and is able to pick up various types of genetic variations from SNVs to small gene deletions, duplications, insertions, or gene rearrangements associated with disease phenotypes. However, advantages include cost-effectiveness and manageable data for clinicians, making clinical decisions easier with more specific disease-relevant information. It can give much deeper coverage up to 5000× for rare alleles in genetic diseases, as well as for low-abundant evolving mutant clones arising as a result of tumor heterogeneity or disease evolution in cancer [38]. The candidate gene approach or commercially available targeted panels is the result of WGS/WES projects carried out at the population scale. The germline, as well as somatic variants, can be tested using targeted NGS panels, few examples of which are listed in Table 2. Targeted panels work on a simple approach of enrichment by amplification using pools of region-specific oligonucleotide primers. Specific size libraries that are produced are then sequenced and analyzed bioinformatically [39].

### 3.2. Transcriptomics

Next-generation sequencing (NGS) has had a transformative impact on transcriptomics, revolutionizing our ability to study the transcriptome—the complete set of RNA molecules in an organism or specific cell population. NGS technologies offer high-throughput and cost-effective methods for profiling and analyzing RNA molecules, allowing researchers to gain deep insights into gene expression, alternative splicing, non-coding RNA regulation, and various biological processes and diseases [40,41,42,43]. Here are some key roles of NGS in transcriptomics:(a)mRNA Sequencing (RNA-Seq): RNA-seq is a widely used NGS application in transcriptomics. It involves the sequencing and quantification of mRNA molecules, providing a comprehensive snapshot of the expressed genes in a biological sample. By generating millions of short sequencing reads, NGS allows researchers to identify and quantify gene expression levels accurately. RNA-seq data can be analyzed to detect differential gene expression between different conditions, discover novel transcripts, assess alternative splicing events, and study gene expression dynamics over time or across different tissues or cell types [44,45].(b)Alternative Splicing Analysis: Alternative splicing, a process in which a single gene can generate multiple mRNA isoforms, significantly contributes to transcriptome complexity. NGS provides the ability to study alternative splicing patterns comprehensively. By aligning RNA-seq reads to the reference genome, researchers can identify splice junctions and detect alternative splicing events. This information allows for the quantification and characterization of transcript isoforms, providing insights into isoform diversity, tissue-specific expression, and the functional implications of alternative splicing [46].(c)Long Non-Coding RNA (lncRNA) and Small-RNA Analysis: NGS facilitates the study of non-coding RNAs, which play critical roles in gene regulation. Techniques such as small-RNA sequencing and long non-coding RNA sequencing enable the identification and characterization of various classes of non-coding RNAs. Small-RNA sequencing allows the profiling of small regulatory RNAs, including microRNAs, piRNAs, and snoRNAs, providing insights into their roles in post-transcriptional gene regulation. Long non-coding RNA sequencing enables the identification and analysis of long non-coding RNA transcripts, which have been implicated in diverse biological processes and diseases [47,48,49]. Long RNA-seq reads can inform about the connectivity between multiple exons and reveal sequence variations (SNPs) in the transcribed region [50]. Small-RNA sequencing is a non-targeted approach that allows the detection of novel miRNA and other small RNAs [51]. The transcriptome with ChIP-seq studies in cancer biology has helped to understand the emerging role of ncRNAs such as sncRNAs and lncRNA in gene regulation mechanisms during carcinogenesis/cancer progression [52,53,54].(d)Transcriptome Assembly and Annotation: NGS data can be utilized to reconstruct and annotate the transcriptome of an organism. By aligning RNA-seq reads to a reference genome or using de novo assembly approaches, researchers can identify novel transcripts, splice variants, untranslated regions, and other transcript features. This information enhances our understanding of the transcriptome’s complexity and improves the annotation of reference genomes, enabling the discovery of previously unknown genes and regulatory elements [55].(e)Single-Cell Transcriptomics: NGS has facilitated the emergence of single-cell transcriptomics, enabling the study of gene expression profiles at the individual cell level. Single-cell RNA-seq (scRNA-Seq) technologies allow the profiling of transcriptomes from individual cells, providing insights into cellular heterogeneity, cell type identification, cell lineage analysis, and gene expression dynamics in complex tissues or developmental processes [56,57].(f)Integrative Transcriptomics: NGS data from transcriptomics can be integrated with other omics data, such as genomics, epigenomics, and proteomics, to gain a comprehensive understanding of gene regulation and biological processes. Integrative approaches provide a system-level view of molecular interactions and enable the identification of key regulatory mechanisms underlying cellular processes and diseases [56].

### 3.3. Epigenomics

Epigenomics refers to the study of epigenetic modifications, which are heritable changes in gene expression patterns that do not involve alterations in the DNA sequence [58,59]. The most common types of epigenetic modifications studied are DNA methylation [60], histone modification, and RNA methylation (epi-transcriptome). These chemical tags in turn alter DNA accessibility, chromatin remodeling, and nucleosome positioning [61]. These modifications are influenced by environmental factors such as nutrients, pollutants, toxicants, and inflammation [62,63]. The knowledge and data generated through whole-genome-wide sequencing in humans, plants, and animals [64] have helped scientists to gain better insights into these epigenetic alterations, especially DNA methylation and hydroxymethylation. Epigenetic alterations have attracted researchers’ and clinicians’ interest in complex disorders such as behavioral disorders, memory, cancer, autoimmune disease, addiction, neurodegenerative, and psychological disorders [65]. There are various platforms and assays developed to study epigenetic modifications, which have been very well described elsewhere [66]. NGS has been utilized for investigating epigenomics, as discussed below:(a)DNA Methylation Profiling: DNA methylation is a crucial epigenetic modification that plays a critical role in gene regulation and cellular processes. NGS enables genome-wide profiling of DNA methylation patterns at single-nucleotide resolution [67]. Several strategies, such as whole-genome bisulfite sequencing (WGBS) and reduced representation bisulfite sequencing (RRBS), leverage NGS to identify methylated cytosines [68]. However, RRBS is based on enriching methylated genomic regions using restriction enzymatic digestion [66,69]. These methods allow researchers to study DNA methylation dynamics, uncover differentially methylated regions (DMRs) associated with diseases, and understand the impact of methylation on gene expression.(b)Chromatin Accessibility Mapping: NGS-based techniques, such as assay for transposase-accessible chromatin using sequencing (ATAC-seq) and DNase-seq, enable the genome-wide profiling of chromatin accessibility. These methods identify regions of the genome that are accessible to DNA-binding proteins and transcription factors, providing insights into gene regulatory elements, enhancers, and promoters. By combining chromatin accessibility data with other epigenetic modifications, gene expression data, and transcription factor binding data, researchers can unravel the functional elements within the genome [70,71].(c)Histone Modification Analysis: Histone modifications, including acetylation, methylation, phosphorylation, and more, are critical epigenetic marks that regulate chromatin structure and gene expression. Chromatin immunoprecipitation sequencing (ChIP-seq) enables genome-wide profiling of histone modifications by antibody-based pull down of the protein followed by enrichment of DNA bound to the protein and sequencing. This technique finds application in many different areas of research, such as transcription factor (TF) binding site identification, histone modification analysis of the DNA, and DNA methylation. For studying histone modifications, antibodies targeted to histone modifications are used to pull down the DNA and sequenced using the NGS technique. The resulting reads are aligned to the reference genome, enabling the identification of histone modification patterns at specific genomic regions. ChIP-Seq can provide insights into the epigenetic regulation of gene expression, chromatin states, and the identification of enhancers and other regulatory elements [72,73,74,75].(d)Chromatin Conformation Analysis: NGS-based techniques, such as Hi-C and 4C-seq, allow the investigation of 3D chromatin organization and interactions. These methods capture long-range chromatin interactions and enable the construction of chromatin interaction maps [76,77]. By integrating 3D chromatin conformation data with epigenetic modifications, gene expression data, and functional annotations, researchers can gain insights into the spatial organization of the genome and understand how it influences gene regulation.(e)In addition to these standalone approaches, NGS data from epigenomics can be integrated with transcriptomics data to unravel the relationship between epigenetic modifications and gene expression. Integration of DNA methylation profiles with RNA-seq data can identify differentially methylated regions (DMRs) associated with gene expression changes. Integration of histone modification and chromatin accessibility data with RNA-seq allows the identification of regulatory elements associated with specific gene expression patterns and the exploration of epigenetic regulatory mechanisms.

### 3.4. Metagenomics

Metagenomics deals with direct genetic analysis of the prokaryotic genome including bacteria, fungi, and viruses contained in a sample [78] either by targeted approach or adaptor ligation PCR approach for shotgun sequencing in a culture-independent manner. The hypervariable region in 16S or 18S ribosomal RNA genes of bacteria and fungi is used in the targeted approach. A blend of conserved and hypervariable regions helps in the identification of each bacterial species from the sample. Similarly, for fungal species identification, ITS1 and ITS2 regions spanning the 5.8S rRNA gene of the fungal genome are selected for amplification [79]. For viral genome sequencing, reads generated from NGS (shotgun) are again the culture-independent method for studying viral diversity, abundance, and functional potential of viruses in the environment. All filtered reads are mapped with the human reference sequence, and remaining, unmapped reads are mapped against the NCBI RefSeq viral genomic database (Table 3) [80]. The targeted viral and bacterial genome panels are also available, e.g., ChapterDx for HR HPV and microbial infection detection, the HIV drug resistance panel, the AMR panel, the gastrointestinal disorder panel, etc.

Based on the nucleotide sequence similarities, pre-processed sequences are clustered at 97% similarity into operational taxonomic units (OTUs). OTUs are compared with the database to identify the microorganisms [81]. Several analysis pipelines are used for the analysis of 16S amplicon reads (Table 3) [82]. For shotgun metagenomics samples, taxonomic and functional profiles can be obtained by different approaches, as elaborated in Table 3 [83,84,85,86,87,88,89]. Microbiome sequencing can identify the full spectrum of microbial species present in the sample. The results are highly quantitative, and one can study the bacterial communities over a specific interval of conditions. The NGS platform can also generate reads for low-abundance species in a sample.

## 4. Bioinformatic Approaches for NGS Data Analysis

NGS generates vast amounts of DNA or RNA sequences, necessitating computational methods to handle, analyze, and interpret these data. Raw sequencing data produced by NGS instruments need to be processed, analyzed, and interpreted to derive biological insights. This is where bioinformatic approaches come into play. These approaches encompass a wide range of computational methods, algorithms, and tools that handle preprocessing, alignment, variant calling, gene expression quantification, differential expression analysis, and other specialized analyses. Once processed, various computational techniques, such as de novo assembly, reference-based mapping, and transcriptome analysis, are employed to extract meaningful biological information. Furthermore, advanced bioinformatic tools facilitate the identification of genetic variations, including single-nucleotide polymorphisms (SNPs), copy number variations (CNVs), and structural variants. Integrative analyses, combining NGS data with other genomic and functional data sources, enable the exploration of gene expression and regulatory networks. The various bioinformatics tools used in NGS analysis are listed in Table 3.

**Table 3 biology-12-00997-t003:** Bioinformatic steps and tools used for NGS data analysis.

Analysis	Commonly Used Tools
**Common Analysis**	
Quality check of sequences	FastQC [90], FASTX-toolkit [91], MultiQC [92]
Trimming of adaptors and low-quality bases	Trimmomatic [93], Cutadapt [94], fastp [95]
Alignment of sequence reads to reference genome	BWA [96], Bowtie [97], dragMAP [98]
Reports visualization	MultiQC [92]
**Whole-Genome Sequencing/Whole-Exome Sequencing/Targeted Panel**	
Removal of duplicate reads	Picard [99], Sambamba [100]
Variant calling (single-nucleotide polymorphisms and indels)	GATK [101], freeBayes [102], Platypus [103], VarScan [104], DeepVariant [105], Illumina Dragen [106]
Filter and merge variants	bcftools [107]
Variant annotation	ANNOVAR [108], ensemblVEP [109], snpEff [110], NIRVANA [111]
Structural variant calling	DELLY [112], Lumpy [113], Manta [114], GRIDDS [115], Wham [116], Pindel [117]
Copy number variation (CNV) calling	CNVnator [118], GATK gCNV [119], cn.MOPS [120], cnvCapSeq(targeted sequencing) [121], ExomeDepth (CNVs from Exome) [122]
**Transcriptomics**	
Alignment of reads to reference	Splice-aware aligner such as TopHat2 [123], HISAT2 [124], and STAR [125]
Transcript quantification	featureCounts [126], HTSeq-count [127], Salmon [128], Kallisto [129]
Differential gene expression analysis enrichment of gene categories	DESeq2 [130], EdgeR [131], DAVID [132], clusterProfiler [133], Enrichr [134]
**Epigenomics-Methyl Seq**	
Sequence aligners	Bwameth [135], BS-Seeker2 [136], Bismark [137]
Methylation level quantification	MethylDackel *
Differential methylation	Metilene [138], BSsmooth [139], methylKit [140]
**Epigenomics-ChIP seq**	
Removal of PCR duplicates	Samtools [107]
Peak calling	MACS2 [141], SICER2 [142], SPP [143]
Peak filtering	Bedtools [144]
Enrichment quality control	ChipQC [145], Phantompeakqualtools [146]
Enrichment comparison	diffBind [147], MAnorm [148], MMDiff [149]
Motif analysis	MemeCHiP [150], Homer [151], RSAT [152]
**16s rRNA seq**	
16S rRNAseq analysis pipelines	QIIME2 [82], mothur [153], USEARCH [154]
Ribosomal RNA databases	Greengenes [155], Silva [156], RDP [157]
**Shotgun Metagenomics**	
Taxonomic classification	MetaPhlAn4 [158], Kaiju [159], Kraken [160]
Assembly of metagenomic reads	metaSPAdes [86], metaIDBA [87]
Protein databases for taxonomic classification	NCBI non-redundant protein database [83]
Gene annotation	Prokka [88], MetaGeneMark [89]
Databases for functional annotation of genes	COG [161], KEGG [84], GO [85]

Footnote: ANNOVAR—ANNOtate VARiation; BWA—Burrows Wheeler Aligner; cn.mops Copy Number Estimation by a Mixture Of PoissonS; COG—Clusters of Orthologous Groups of Proteins; DAVID—A Database for Annotation, Visualization and Integrated Discovery; Ensembl VEP—Ensembl Variant Effect Predictor; Fastp—Fsatq Preprocessor; GATK—Genome Analysis Tool Kit; GO—Gene Ontology; HISAT2—Hierarchical Indexing for Spliced Alignment of Transcripts; HOMER—Hypergeometric Optimization of Motif EnRichment; Htseq-count—High-Throughput Sequence Analysis in Python; KEGG: Kyoto Encyclopedia of Genes and Genomes; NCBI—National Center for Biotechnology Information; MACS: Model-Based Analysis for ChIP-Seq; MEME—Multiple EM for Motif Elicitation; Meta-IDBA—Meta-Iterative De Bruijn Graph De Novo Short-Read Assembler; MetaPhlAn—Metagenomic Phylogenetic Analysis; metaSPAdes—meta St Petersburg Genome Assembler; QIIME—Quantitative Insights Into Microbial Ecology; RDP—Ribosomal Database Project; RSAT—Regulatory Sequence Analysis tools; SICER—Spatial Clustering Approach for the Identification of ChIP-Enriched regions; SPP—The Signaling Pathways Project; STAR—Spliced Transcripts Alignment to a Reference. * Available at: https://github.com/dpryan79/MethylDackel/ (accessed on 1 June 2023). Bold represents the categories of analysis and commonly used bioinformatics tools used for NGS data analysis.

## 5. NGS Applications in Research and Diagnostics

NGS has revolutionized the field of scientific research and clinical genomics due to high-throughput multiplexing. This power of NGS in translation medicine lies not only in its advanced multiplexing efficiency but also in the equally smart bioinformatic tools used for data curation followed by various reference databases that help researchers, medical practitioners, and drug designers to understand the genetic basis of the disease. Different population genome sequencing projects such as 1000 G, ExAC, ESP6500, UK 100 K, Indigenome, and gnomAD generated vast amounts of data on NGS [162]. Among the reference population databases, gnomAD is the largest and most widely used database generated from harmonized sequencing data incorporating exome and genome sequencing data from 140,000 humans. This has been widely used as a resource for estimating allele frequency in rare diseases, disease gene discovery, and the biological effect of variation [163]. This has led to the creation of knowledge bases and in turn large and small sequencing panels for major applications in clinical research and diagnostics [164]. The large gene panels find their major application in clinical research mainly in cancer genetics.

### 5.1. Role of NGS in Research

#### 5.1.1. Microbiome Research

Given the ubiquitous nature of microbes, their symbiotic, pathogenic, and commensal characteristics are of importance to humans by forming a highly functioning ecosystem. The microbiome community became an obligatory factor in our survival through evolution [165]. However, a close monitoring and comprehensive understanding of the host–microbiome and microbiome–intercommunity interactions are vital to healthy survival. The approaches include pathogen surveillance, functional dysbiosis, and therapeutic potential. Metagenomic studies have linked the gut microbiome to disorders affecting mental health [166], autoimmune diseases (rheumatoid arthritis) [167], and metabolic disorders (diabetes and obesity) [168], thus instrumental in evaluating the functional potential of the microbiome. This opens doors for more therapeutic approaches and options. Designing targeted panels to pick up mutations (aiding in antibiotic resistance tracking) or identifying the pathogenic genes followed by sequencing can help in detecting pathogens with known antimicrobial resistance. Research is also underway for the pharmacomicrobiomics of individuals requiring drug treatment. This would aid in identifying the effect of drugs on an individual’s microbiome and drug disposition by the microbiome.

#### 5.1.2. Human Disease Research

The focus of NGS-based research is now extended from genomic research to the study of transcriptome, epi-transcriptome, and epigenome. Human genome-based research through WGS and WES has provided novel insights into the biological processes and has found application in wellness research; agriculture and food research; genome-wide association research studies uncovering the wide range of population genetic variants; their genetic linkage and molecular basis to various diseases, including cancer; and the study of new pathogenic/emerging variants such as SARS-CoV-2 variants in human diseases. The redefinition of the mutational landscapes in tumors has resulted in translating this information into clinical research through the ever-growing list of targeted large gene panels such as the 261 gene panel, the 400 gene panel, the TSO 500 panel from Illumina, IDT, Agilent, and Thermo Fisher. These panels assess not only SNVs but also clinically relevant CNVs and RNA fusion transcripts, TMB, and microsatellite instability (MSI) for lung cancer, breast cancer, colorectal cancer, and even for difficult cancers such as ovarian, pancreatic, renal, urothelial cancers, etc.

RNA-seq finds its application mainly in research for analyzing pathogen transcriptomic signatures [169], metastatic biomarkers, therapeutic resistance, immune microenvironment, immunotherapy, and neoantigen research in cancer [170,171]. With NGS, it is now possible to study single-cell behavior with respect to its differentiation, de-differentiation, proliferation, and tumorigenesis in cancer using single-cell RNA-sequencing strategies such as Smart-seq2, MATQ-seq, SUPeR-seq, Drop-seq, Seq-Well, Chromium, DroNC-seq, STRT-seq, etc. [172]. The recent new development of the RiboSeq technique can plot potential ongoing events of translation in the cytosol, which is useful in identifying potentially functional micro-peptides. This is how thousands of sORFs (small open-reading frames) were discovered in lncRNA. Thus, with transcriptomics, Ribo-seq, and MS proteomics, the bifunctional potential of RNA molecules is identified [173,174].

The role of epigenomics in gene regulation, the maintenance of tissue-specific expression, and developmental processes is evident from X chromosome inactivation, embryonic development, genomic imprinting, epigenetic reprogramming, cell identity establishment, and lineage specification studies. Epigenetic signatures are important biomarkers that have promise not only in cancer, malignant transformation, and metastasis but also for their clinical applicability in other disease conditions such as diabetes, neurological conditions, infectious diseases, and immune disorders [175,176]. The reversible nature of epigenetic changes makes them promising candidates for precision medicine in cancer and other conditions [164,176]. Pharmacoepigenomics is an emerging research area, where the relationship between variable drug response and epigenetic status is being studied [59]. Epi-drugs have been developed over the last 40 years, and few are in clinical practice, whereas some are in clinical trials [177]. Non-coding RNAs (ncRNAs) are gene expression regulators apart from epigenetic modifications that are being explored as drug targets. Numerous lncRNAs are subsequently identified and found to be aberrantly expressed in various tumors [58]. Increasing studies have shown miRNAs as biomarkers of multiple cancers as their abnormal quantity has been correlated with the stage of pathology and prognosis [178]. The applications of miRNA analog or anti-miRNAs have shown promising outcomes in vitro and in vivo cancer studies, suggesting that miRNA-based drugs are emerging as a novel strategy for cancer therapy [179]. Apart from cancer, multiple FDA-approved drugs exist for DMD, SMA, familial hypercholesterolemia, CMV retinitis, etc. [178].

### 5.2. NGS in Diagnostics

A decisive approach is important when selecting an NGS assay. Type of variant, disease symptoms, and probable genetic associations are important aspects when selecting NGS-based tests in clinical decision making, as per recommendations by the National Comprehensive Cancer Network (NCCN), the College of American Pathologists (CAP), the American Society of Clinical Oncology (ASCO), the Association of Molecular Pathology (AMP), the American College of Medical Genetics (ACMG), and the European Society of Medical Oncology (ESMO).

#### 5.2.1. Infectious Diseases

The identification of the exact etiological agent in microbial infections is important for precision medicine, which has driven the approach of syndromic testing/multiple pathogen testing assays such as BioFire or multiplex PCRs. However, with the limitations of multiplexing, NGS panels are being developed that can detect any pathogen using a shotgun approach or a targeted approach (16S) from various diseased specimens or clinical isolates. These panels can not only pick up causative pathogens but can be used to identify drug-resistant mutations such as antimicrobial drug-resistant mutations and antiviral drug-resistant mutations [180]. The useful data generated through NGS on microbial identification and drug resistance genotyping, e.g., in MTB, HIV, and SARS-CoV-2 [181], have proven important for disease surveillance, disease containment, public health epidemiological studies, policy making, and rapid therapeutic interventions, as evident during the COVID-19 outbreak [182]. However, with the need for fast diagnosis, NGS, in its current form for infectious pathogen detection, cannot replace current standard point-of-care testing such as PCR, multiplex BioFire panel testing, or multiplex QPCR commercial kits.

#### 5.2.2. Inherited Genetic Diseases

The association of multiple genes in multifactorial disorders such as diabetes, hypercholesterolemia, infertility, etc., has been discovered in the rapidly emerging field of genomics. For example, the classical approach to comprehending the genes participating in infertility, gametogenesis, the hormonal cycle, fecundation, and embryo development would have been difficult and time-consuming. Targeted NGS panels have evolved as a result of WGS, and WES has enabled the simultaneous evaluation of multiple genes and their variants explaining the complexity of various disorders, including infertility, inherited genetic diseases, and reproductive genome testing, including NIPT (non-invasive prenatal testing), PGS/PGD (preimplantation genetic disease testing), and pediatric disorders such as developmental delay disorders, metabolic syndromes [183]. This has enabled disease treatment through personalized genome testing for the betterment of human health, preventive testing, and disease management.

#### 5.2.3. HLA Typing

NGS-based HLA typing using WGS or targeted panels over conventional HLA typing methods for organ transplant or HSCT provides more unambiguous, high-throughput, high-resolution typing results from a single platform. This approach provides complete information on all the HLA loci involved in (1) the etiopathogenesis of immune disorders such as coeliac disease, psoriasis, rheumatoid arthritis, type I diabetes, SLE, lung diseases (e.g., asthma or sarcoidosis) [184], infectious disease predispositions (e.g., HIV, hepatitis, leprosy, tuberculosis), and other conditions such as malignancies and neuropathies [185]) generating population/ancestry-based database.

Epigenetics study through methylation profiling was in fact first studied using the HLA gene, which has its epigenetic regulators located in the non-coding region such as enhancers, promoters, and UTR regions that regulate HLA gene expression. Bioinformatically, the sequence data obtained are analyzed using commercial HLA-specific software such as NGSengine or exome-data-based software such as OptiType [186], Polysolver [187], xHLA [188], and HLAminer [189] to determine the HLA types [190].

#### 5.2.4. Cancer

The comprehensive human genome sequencing project, WGS and WES, has identified cancer as the disease of the genome and is a multifactorial disease with non-mendelian (Somatic) origin in the majority of cases and mendelian origin in inherited cancers. Through the efforts of TCGA (The Cancer Genome Atlas) and ICGC (International Cancer Genome Consortium), the understanding of cancer and the comprehensive gene alteration data in protein-coding regions for all types of human cancers are now readily available [191].

Different enterprises, such as FoundationOne by Foundation Medicine (Cambridge, MA, USA), Oncomine by Thermo Fisher (Waltham, MA, USA), CANCERPLEX by KEW (Cambridge, MA, USA), MSK-IMPACT by the Memorial Sloan Kettering Cancer Center (New York, NY, USA), OmniSeq Advance by the Roswell Park Cancer Institute (Buffalo, NY, USA), the CC Onco Panel by Sysmex (Kobe, Japan), and the Todai Onco Panel by Riken Genesis (Tokyo, Japan) have come up with multigene panels using TCGA and ICGC data for different NGS platforms that are now frequently used in cancer prognosis and therapeutics [191]. Figure 4 summarizes the various data integration methods for cancer diagnosis, prognosis, and therapeutics [192]. Though all alterations picked up in NGS may not find immediate application in translation medicine, they help discover the different pathways operating in cancer pathogenesis and build on the cancer genomics database. Lung cancer biomarkers have been developed for almost over a decade for the development of a commercial NGS panel of 15–21 genes for precision oncology in lung cancer, picking up all types of structural variants (SVs) on a single platform [193,194]. This landmark study of precision oncology in lung cancer opened the doors for various solid tumors such as CRC, breast, ovarian, endometrial, pancreatic, and even liquid tumors such as myeloid and lymphoid malignancies to use NGS panels effectively with limited sample requirement, infrastructure, and different technical and analytical expertise [98]. Thus, a comprehensive gene testing approach in cancer provides maximum treatment efficacy and reduces the window period of disease progression in a cancer patient, resulting in improved QOL (quality of life), PFS (progression-free survival), and OS (overall survival).

One important aspect of somatic mutation testing in cancer is tumor heterogeneity. It needs to be clearly and carefully dealt with by setting the variant calling cutoff thresholds to avoid false-positive or false-negative variant calling and reporting [195]. Being the most sensitive method of mutation detection, evolving mutant clones, the allelic burden of mutation and thus the disease progression can be determined through NGS. Liquid biopsy testing in cancer has become a very handy tool in tracking disease progression and treatment monitoring in clinical oncology using the circulating tumor DNA in a metastatic setting [196]. NGS plays a crucial role in identifying biomarkers associated with hereditary/germline cancers. For example, in the case of hereditary breast and ovarian cancer syndrome (HBOC), the understanding of its genetic basis has evolved beyond the BRCA1 and BRCA2 mutations. The inclusion of other genes involved in the homologous recombination repair (HRR) pathway, known as BRACAness genes, has reshaped our understanding of HBOC. These additional genes include CDH1, PTEN, TP53, STK11, PALB2, ATM, CHEK2, MUTYH, BARD1, MRE11A, NBN, RAD50, RAD51C, RAD51D, and NF1, in addition to BRCA1 and BRCA2. NGS has facilitated the identification and characterization of these extended sets of genes associated with HBOC, expanding our knowledge of hereditary cancer predisposition [197].

### 5.3. NGS in Forensics

Ever since 1984, when Sir Alec Jeffreys first proposed the application of DNA profiling to distinguish between different samples at a crime site, DNA analysis has emerged as a prime investigative tool in forensic science [198]. This field is now being dominated by NGS, keeping behind the old methods of DNA fingerprinting such as restriction fragment length polymorphism (RFLP), mitochondrial DNA, variable number of tandem repeat (VNTR) profiling, and short tandem repeat (STR) typing to solve an array of criminal mysteries [199]. NGS has gained rapid importance in this domain due to its ability to deliver highly accurate, reproducible, and results of the highest sensitivity from highly contaminated and degraded sample qualities received in forensic labs [200]. NGS is being applied to solve different categories of criminal cases: mtDNA for the investigation of maternal lineage [201], Y chromosome STR analysis for the identification of male DNA in a contaminated sample [202], animal and plant DNA analysis to identify important clues in poisoning cases [203], ancestry tracing [204], predicting phenotypes based on the genes [205], epigenetic analysis to identify the age of the donor DNA [206], and microRNA analysis for identifying body fluids and post-mortem interval [207]. The application of NGS in biodefense and bioterrorism involving the detection of microbial signatures at crime sites is another discipline gaining rapid attraction [208,209]. The major providers of NGS technology dominating the forensic domain are Illumina’s MiSeq FGx, Thermo Fisher’s Ion Torrent PGM, and Ion S5 [210,211]

## 6. Future Prospects and Conclusions

The future scope of NGS holds tremendous potential for advancements and applications in various fields. The progress in bioinformatics, robotics, liquid handling, and nucleic acid preparation will revolutionize NGS sequencing methods, making them faster and more precise. These forthcoming sequencing platforms will necessitate smaller amounts of input DNA and reagents, scaling down to zeptoliters and even a few molecules. Additionally, they will become increasingly portable, enabling their utilization in diagnostic applications across various fields such as medical, agricultural, ecological, and other field-based settings. Taken together, NGS holds immense potential for transformative advancements across multiple domains. NGS has already revolutionized fields such as clinical diagnostics, cancer genomics, and microbial genomics, providing unprecedented insights into the genetic underpinnings of diseases and driving personalized medicine. As technology progresses, NGS is expected to play a pivotal role in areas such as single-cell genomics, long-read sequencing, epigenomics, and multi-omics integration, enabling a deeper understanding of cellular processes, disease mechanisms, and personalized treatment strategies. The development of real-time sequencing and point-of-care applications will further extend the reach of NGS, empowering rapid diagnostics and monitoring in various settings. Additionally, advancements in bioinformatics and data analysis will be crucial for extracting meaningful insights from the vast amount of NGS data generated. The higher order multiplexing will enable more samples to be processed in a shorter time and at a reduced cost supported by the advances in robotics, liquid handling, and sample processing will contribute to these advancements. Equally important will be advanced in faster and more accurate bioinformatic data analysis, as well as data transfer and storage. With ongoing technological improvements and cost reduction, NGS will become more accessible and widespread, facilitating its integration into routine clinical practice, research, agriculture, and environmental studies. The future of NGS is promising, promising to unlock new frontiers of knowledge and catalyze advancements that will have a profound impact on human health, agriculture, environmental conservation, and beyond.

## Figures and Tables

**Figure 1 biology-12-00997-f001:**
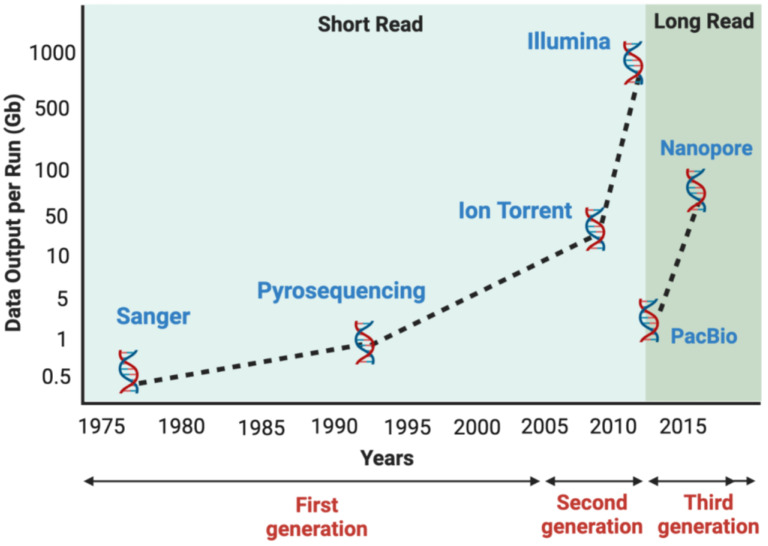
Evolution of sequencing technologies. The development of sequencing technologies over the past four decades can be categorized into three generations. The first generation was represented by Sanger sequencing, providing the foundation for DNA sequencing. The second generation introduced massively parallel sequencing with platforms such as Illumina and Ion Torrent, enabling high-throughput sequencing. The current third generation includes PacBio and Nanopore, offering long-read and single-molecule sequencing capabilities.

**Figure 2 biology-12-00997-f002:**
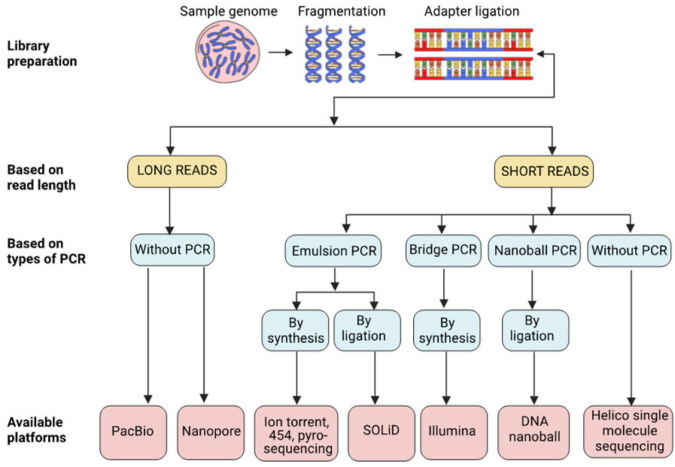
Overview of various NGS technologies with different platforms and principles.

**Figure 3 biology-12-00997-f003:**
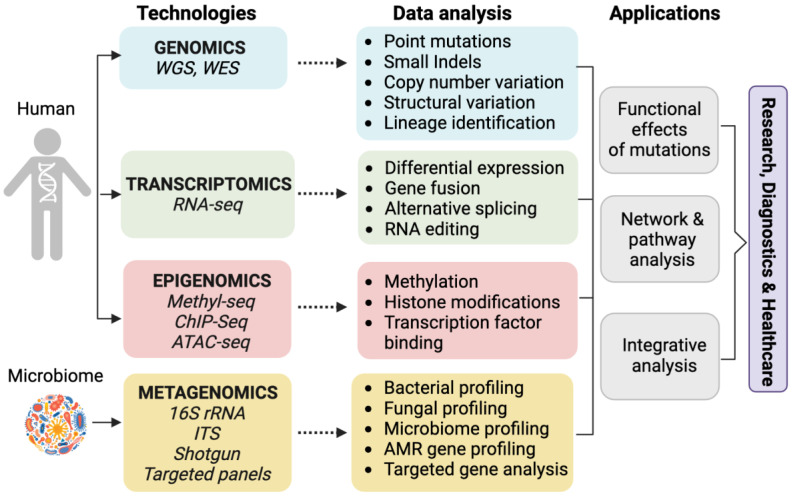
Various approaches used for genome analysis and applications of NGS, including technological platforms, data analysis, and applications. WGS, whole-genome sequencing; WES, whole-exome sequencing; Seq, sequencing; ITS, internal transcribed spacer; ChIP, chromatin immunoprecipitation; ATAC, assay for transposase-accessible chromatin; AMR, anti-microbial resistance.

**Figure 4 biology-12-00997-f004:**
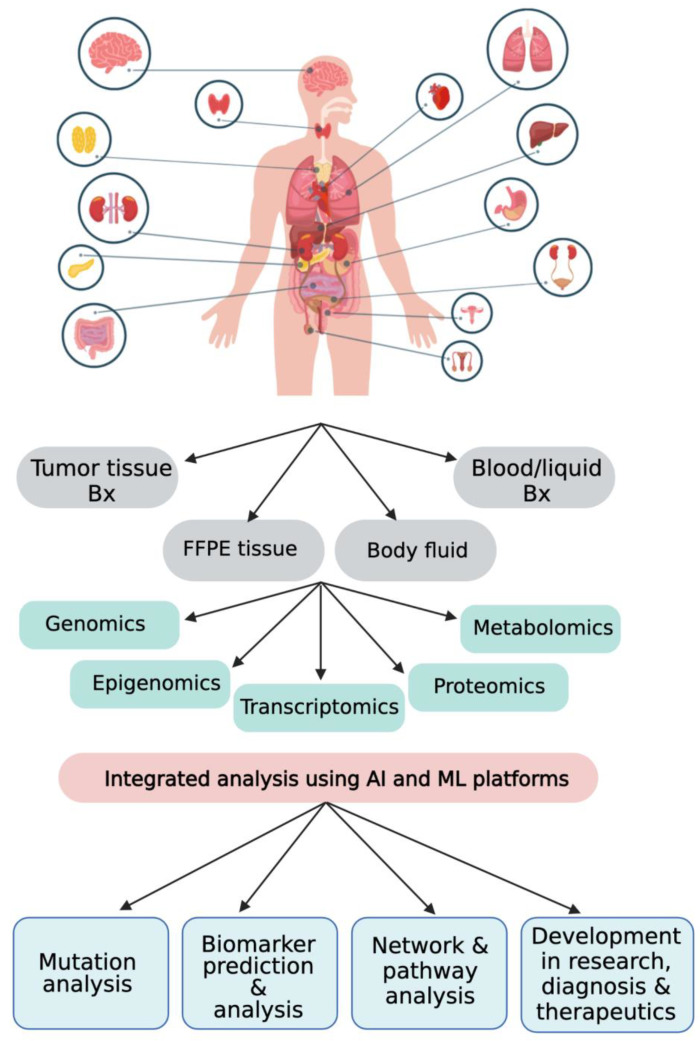
Role of NGS technology in cancer diagnosis, prognosis, and therapeutics using an integrative omics approach. FFPE, formalin-fixed paraffin-embedded; Bx, biopsy; AI, artificial intelligence; Ml, machine learning.

**Table 1 biology-12-00997-t001:** Different generations of NGS platforms.

Sr No.	Platform	Use	Sequencing Technology	Amplification Type	Principle	Read Length (bp)	Limitations	Ref.
1	454 pyrosequencing	Short read sequencing	Seq by synthesis	Emulsion PCR	Detection of pyrophosphate released during nucleotide incorporation.	400–1000	May contain deletion and insertion sequencing errors due to inefficient determination of homopolymer length.	[18,19,20]
2	Ion Torrent	Short read sequencing	Seq by synthesis	Emulsion PCR	Ion semiconductor sequencing principle detecting H^+^ ion generated during nucleotide incorporation.	200–400	When homopolymer sequences are sequenced, it may lead to loss in signal strength.	[19,20,21]
3	Illumina	Short read sequencing	Seq by synthesis	Bridge PCR	Solid-phase sequencing on immobilized surface leveraging clonal array formation using proprietary reversible terminator technology for rapid and accurate large-scale sequencing using single labeled dNTPs, which is added to the nucleic acid chain.	36–300	In case of sample overloading, the sequencing may result in overcrowding or overlapping signals, thus spiking the error rate up to 1%.	[19,20,22]
4	SOLiD	Short read sequencing	Seq by ligation	Emulsion PCR	An enzymatic method of sequencing using DNA ligase. 8-Mer probes with a hydroxyl group at 3′ end and a fluorescent tag (unique to each base A, T, G, C) at 5′ end are used in ligation reaction.	75	This platform displays substitution errors and may also under-represent GC-rich regions. Their short reads also limit their wider applications.	[20,23]
5	DNA nanoball sequencing	Short read sequencing	Seq by ligation	Amplification by Nanoball PCR	Splint oligo hybridization with post-PCR amplicon from libraries helps in the formation of circles. This circular ssDNA acts as the DNA template to generate a long string of DNA that self-assembles into a tight DNA nanoball. These are added to the aminosilane (positively charged)-coated flow cell to allow patterned binding of the DNA nanoballs. The fluorescently tagged bases are incorporated into the DNA strand, and the release of the fluorescent tag is captured using imaging techniques.	50–150	Multiple PCR cycles are needed with a more exhaustive workflow. This, combined with the output of short-read sequencing, can be a possible limitation.	[24,25]
6	Helicos single-molecule sequencing	Short-read sequencing	Seq by synthesis	Without Amplification	Poly-A-tailed short 100–200 bp fragmented genomic DNA is sequenced on poly-T oligo-coated flow cells using fluorescently labeled 4 dNTPS. The signal released upon adding each nucleotide is captured.	35	Highly sensitive instrumentation required. As the sequence length increases, the percentage of strands that can be utilized decreases.	[26,27]
7	PacBio Onso system	Short-read sequencing	Seq by binding	Optional PCR	Sequencing by binding (SBB) chemistry uses native nucleotides and scarless incorporation under optimized conditions for binding and extension (https://www.pacb.com/technology/sequencing-by-binding/, accessed on 1 July 2023).	100–200	The higher cost compared to other sequencing platforms.	
8	PacBio Single-molecule real-time sequencing (SMRT) technology	Long-read sequencing	Seq by synthesis	Without PCR	The SMRT sequencing employs SMRT Cell, housing numerous small wells known as zero-mode waveguides (ZMWs). Individual DNA molecules are immobilized within these wells, emitting light as the polymerase incorporates each nucleotide, allowing real-time measurement of nucleotide incorporation	average 10,000–25,000	The higher cost compared to other sequencing platforms.	[28,29]
9	Nanopore DNA sequencing	Long-read sequencing	Sequence detection through electrical impedance	Without PCR	The method relies on the linearization of DNA or RNA molecules and their capability to move through a biological pore called “nanopores”, which are eight nanometers wide. Electrophoretic mobility allows the passage of linear nucleic acid strand, which in turn is capable of generating a current signal.	average 10,000–30,000	The error rate can spike up to 15%, especially with low-complexity sequences. Compared to short-read sequencers, it has a lower read accuracy.	[5,19,30]

**Table 2 biology-12-00997-t002:** Examples of targeted panels available in research and diagnostic settings.

Disease Condition	Available Panel	Type of Inheritance	Specimen Type
Inherited cardiovascular defects	Cardiovascular research panel	Germline	Blood
Arrhythmias and cardiomyopathies	Arrhythmias and cardiomyopathy research panel	Germline	Blood
Sensitivity to pharmacological drugs	Pharmacogenomics research panel (PGex Seq panel)	Germline	Blood
Antimicrobial treatment efficacy testing	Antimicrobial resistance research panel	Microbial gene testing	Bacterial culture
Infertility conditions	Infertility research panel	Germline	Blood
Homologous recombination defect analysis	HRR gene panel	Somatic	Tumor tissue
myeloid cancers	Myeloid cancer panel	Somatic	Blood
HIV speciation and drug resistance	HIV-Xgene panel	Pathogen detection	HIV-positive plasma
Antimicrobial resistance in MTB	TB research panel	Pathogen detection	MTB-positive specimen
Inborn errors of metabolism	Error of metabolism research panel	Germline	DBS/blood
Hereditary cancers	BRACA and extended breast and ovarian cancer research panel, inherited cancer research panel	Germline	Blood

## Data Availability

Not applicable.

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
