# Peer review of "Next-Generation Sequencing Technology: Current Trends and Advancements"

_biology, 2023, doi:10.3390/biology12070997_

Round 1
Reviewer 1 Report
This review from Satam et. al., is well written and exhaustive covering all aspects of next-generation sequencing including various software tools used in data curation, and its overwhelming applications in cancer diagnostics, organ transplantation, and in infectious diseases. Considering the continuous evolution, especially in relation to development of advanced tools for long-read NGS performance and analysis, and the broad readership in the NGS field, this review might benefit if the authors include a table or paragraph highlighting the advantages and disadvantages of long-read over short-read sequencing.
Author Response
Authors’ Response: We agree with the reviewer’s insightful suggestion regarding the inclusion of a table or paragraph that highlights the advantages and disadvantages of long-read sequencing over short-read sequencing. The relevant paragraph has been included in the revised manuscript.
Reviewer 2 Report
Review article “Next-Generation Sequencing Technology: Current Trends and Advancements” by Satam et al., broadly elaborates NGS technology and its applications. The authors have done good work in compiling and summarizing the data. Few minor concerns need to be addressed.
Minor comments:
1. The authors should provide brief legends for Figs. 3, and 4 to assist readers in understanding the content of the figures.
2. Abbreviations used in Fig.-4, such as "FFPE," "AI," and "ML," should be defined within the figure legend. This will enable all readers, including those unfamiliar with the terminology, to interpret the figures.
3. The authors have discussed various NGS technologies and their potential applications. However, they should also elaborate on the limitations of these techniques. An example of it is existing difficulties in structural variation analysis. Difficulty in repeat region capture and analysis by short reads sequencing.
4. The authors should elaborate on the average length of various reads by different NGS technologies.
5. The author should also discuss the gnomAD database in NGS applications in research and diagnostics.
Author Response
Review article “Next-Generation Sequencing Technology: Current Trends and Advancements” by Satam et al., broadly elaborates NGS technology and its applications. The authors have done good work in compiling and summarizing the data. Few minor concerns need to be addressed.
Minor comments:
- The authors should provide brief legends for Figs. 3, and 4 to assist readers in understanding the content of the figures.
Authors’ Response: As suggested by the reviewer, the figure 3 and 4 legends have been modified in the revised manuscript.
- Abbreviations used in Fig.-4, such as "FFPE," "AI," and "ML," should be defined within the figure legend. This will enable all readers, including those unfamiliar with the terminology, to interpret the figures.
Authors' Response: Abbreviations in Fig. 4, have been defined in the revised manuscript.
- The authors have discussed various NGS technologies and their potential applications. However, they should also elaborate on the limitations of these techniques. An example of it is existing difficulties in structural variation analysis. Difficulty in repeat region capture and analysis by short reads sequencing.
Authors' Response: As per the reviewer's suggestions, the limitations of the NGS platforms have been included in Table-1
- The authors should elaborate on the average length of various reads by different NGS technologies.
Authors' Response: The information related to the read length of sequencing platforms has been included in a separate column in Table-1
Reviewer 3 Report
The review manuscript entitled " Next-Generation Sequencing Technology: Current Trends and Advancements” by Satam et al covers all aspects of the sequencing technologies (genesis to most modern sequencing technology). The authors have put enormous efforts to put together the information about the development of the technologies as well as their applications in various fields. It is a nicely written manuscript and will be helpful for the readers in most of the fields. However, texts in the boxes are not properly readable because of background color in many figures. So, authors should choose appropriate background color for their figures.
Author Response
Authors' Response: Thank you for the suggestion. We have modified the figures color patterns and text to enhance its visibility and readability in the revised manuscript.
Reviewer 4 Report
Satam et al have written a detailed overview about the current state of the art in NGS and its applications. The review is quite compact although filled with many details.
I have some points, which need to be address or might be added:
1. Figure 1: Max output of Illumina MiSeq with V3 is 15Gb the graph shows 1000Gb.
2. Line 94-100: I do not understand the enumeration in the text here. In what context is it placed? Historical? Number of devices in use?
3. Table 1: PacBio’s short-read sequencing by binding technology is missing, with Q40 of >=90%.
4. Targeted sequencing can also be performed by capturing see: https://www.illumina.com/techniques/sequencing/dna-sequencing/targeted-resequencing/targeted-panels.html
5. Line 299: To my knowledge, ChIP-seq is defined as investigation of protein-DNA interaction by sequencing. Primarily by investigation of transcription factors. Histone modification analysis is one application of ChIP-seq.
6. Line 348: In regard of the title of this review: I think it is worth to mention Illuminas Dragen platform, which reduced the computing time for a 30x human whole genome alignment to 20 min.
https://support.illumina.com/content/dam/illumina-support/documents/documentation/software_documentation/dragen-bio-it/dragen-bio-it-platform-v3.2.8-user-guide-1000000085871-00.pdf
7. Table 3: Some algorithms are worth to add:
Quality check of sequences: multiQC
Alignment of sequences reads to reference: dragMAP
Variant calling: DeepVariant
Variant annotation: NIRVANA
Structural Variant: Manta, GRIDDS, Wham
CNV calling: cnvCapSeq
Transcript Quantificaiton/Differential gene expression: Salmon
8. Line 376: gnomAD is missing.
9. Lines 470, 484, 500, 550: Check chapter numbering.
10. All Science references e.g. line 653 have year 1979.
Author Response
Satam et al have written a detailed overview about the current state of the art in NGS and its applications. The review is quite compact although filled with many details.
I have some points, which need to be address or might be added:
- Figure 1: Max output of Illumina MiSeq with V3 is 15Gb the graph shows 1000Gb.
Authors' response: As per the reviewer's suggestion, the figure-1 has been updated in the revised version.
- Line 94-100: I do not understand the enumeration in the text here. In what context is it placed? Historical? Number of devices in use?
Authors' response: The original text was intended to provide a brief overview of the available technologies. However, as the reviewer correctly identified a possibility of confusion, the content has been rewritten for clarity in the revised manuscript.
- Table 1: PacBio’s short-read sequencing by binding technology is missing, with Q40 of >=90%.
Authors’ response: As per the reviewer's valuable suggestion, PacBio's new sequencer (based on Sequencing by binding) has been incorporated in the table-1. For Illumina targeted panels, these have been covered in Table 2 and also cited in the section "5.2.1 Infectious diseases."
- Targeted sequencing can also be performed by capturing see: https://www.illumina.com/techniques/sequencing/dna-sequencing/targeted-resequencing/targeted-panels.html
- Line 299: To my knowledge, ChIP-seq is defined as investigation of protein-DNA interaction by sequencing. Primarily by investigation of transcription factors. Histone modification analysis is one application of ChIP-seq.
Authors’ Response: We thank reviewer for the suggestions. ChIP-seq is a technique to sequence DNA that is enriched after antibody-based immunoprecipitation of specific proteins. The ChIP-seq technique has many applications including identification of transcription factor binding site, DNA methylation analysis and histone modification analysis on DNA. We have included these additional points in the revised manuscript.
- Line 348: In regard of the title of this review: I think it is worth to mention Illuminas Dragen platform, which reduced the computing time for a 30x human whole genome alignment to 20 min.
Authors’ Response: Included Dragen platform in table-3
https://support.illumina.com/content/dam/illumina-support/documents/documentation/software_documentation/dragen-bio-it/dragen-bio-it-platform-v3.2.8-user-guide-1000000085871-00.pdf
- Table 3: Some algorithms are worth to add:
Quality check of sequences: multiQC
Alignment of sequences reads to reference: dragMAP
Variant calling: DeepVariant
Variant annotation: NIRVANA
Structural Variant: Manta, GRIDDS, Wham
CNV calling: cnvCapSeq
Transcript Quantificaiton/Differential gene expression: Salmon
Authors’ response: As per the reviewer's suggestions, these tools have been included in Table-3 of the revised manuscript.
- Line 376: gnomAD is missing.
Authors’ response: As per the reviewer's suggestions, we have included a description on gnomAD, in the revised manuscript.
- Lines 470, 484, 500, 550: Check chapter numbering.
Authors’ response: We thank Reviewer for pointing this out. In the revised version, this error has been corrected
- All Science references e.g., line 653 have year 1979.
Authors’ response: We thank Reviewer for pointing out this typo error. In the revised version, this error has been corrected.